# Factors Associated with Complicated Parapneumonic Pleural Effusion/Empyema in Patients with Community-Acquired Pneumonia: The EMPIR Study

**DOI:** 10.3390/jcm14051739

**Published:** 2025-03-05

**Authors:** Rosa María Bravo Jover, Vicente F. Gil-Guillen, Carlos Pérez Barba, Jose Antonio Quesada, María García López, Isabel García Soriano, María de los Reyes Pascual Pérez

**Affiliations:** 1Internal Medicine Service, Elda General University Hospital, 03600 Elda, Spain; cperezb@coma.es (C.P.B.); isabel.garcia.soriano13@gmail.com (I.G.S.); 2Clinical Medicine Department, University Miguel Hernández de Elche, 03550 Alicante, Spain; vte.gil@gmail.com (V.F.G.-G.); jquesada@umh.es (J.A.Q.); 3Research Unit, Elda General University Hospital, 03600 Elda, Spain; 4Network for Research on Chronicity, Primary Care and Health Promotion (RICAPPS), 03550 San Juan de Alicante, Spain; 5Internal Medicine Service, Orihuela Hospital, 03300 Orihuela, Spain; rbravojover@gmail.com

**Keywords:** empyema, complicated parapneumonic pleural effusion, community-acquired pneumonia

## Abstract

**Objectives:** To identify factors associated with complicated parapneumonic pleural effusion/empyema (CPPE/empyema) in inpatients with community-acquired pneumonia (CAP) and to build a mathematical model for CPPE/empyema. **Methods**: This is an observational case–control study nested within a retrospective cohort, based on clinical practice, and including adults hospitalized with CAP from 2009 to 2019. Cases and controls were defined according to diagnosis of CPPE/empyema during admission. For each case, two controls were randomly selected and matched for the period of admission to avoid seasonality bias. Explanatory variables included demographic, analytical, clinical, and radiological data; treatment with corticosteroids on admission; prognostic and CAP severity scales; comorbidity; and the interval between symptoms onset and admission. **Results**: Of 4372 pneumonias reviewed, 2015 were excluded due to pleural effusion, blunting of the costophrenic angle without thoracentesis, or heart failure. Of the remaining 2357 patients, 106 developed CPPE/empyema (cases), and 212 were selected as controls. Factors associated with CPPE/empyema were pleuritic pain (odds ratio [OR] 7.42, 95% confidence interval [CI] 3.83–14.38), multilobar radiological involvement (OR 4.48, 95% CI 2.26–8.88), and leukocytosis (OR 4.12, 95% CI 1.94–8.76). Corticosteroids showed a protective effect (OR 0.24, 95% CI 0.09–0.61). Age (OR 0.99, 95% CI 0.97–1.02; *p* = 0.56) and sex (OR 1.91, 95% CI 0.94–3.88; *p* = 0.074) were adjustment variables. The area under the receiver operating characteristic curve was 0.847 (95% CI 0.772–0.921). **Conclusions:** Pleuritic pain, multilobar radiological involvement, and leukocytosis are associated with CPPE/empyema in inpatients with CAP. Treatment with corticosteroids upon admission seems to be a protective factor. The discriminative capacity of the resulting multivariable model presents moderate/high accuracy.

## 1. Introduction

Community-acquired pneumonia (CAP) continues to be a significant medical problem and is one of the most prevalent pathologies in hospitalized patients. About 40% of inpatients with pneumonia will develop parapneumonic pleural effusion (PPE) [1,2]. Thoracic empyema, characterized by the presence of purulent fluid in the pleural space, remains an important clinical problem with considerable morbidity and mortality. Complicated PPE (CPPE) appears in the context of CAP with bacterial or other microbial infections, and it requires a thoracostomy tube for resolution. Characteristic findings include an exudative effusion with a high white blood cell count, pH < 7.20, glucose < 2.2 mmol/L (<40 mg/dL), and lactate dehydrogenase (LDH) > 1000 IU/L [3]. When frank pus is found in the pleural space, the clinical entity is known as empyema [3,4,5] (Table 1). The characteristics of pleural fluid may vary in different fluid pockets/loculations.

Although the overall incidence of CPPE/empyema decreased significantly with the advent of effective antibiotic treatments, epidemiological studies suggest that this trend is slowly reversing [6,7]. Despite major advances in the field of antibiotics and new surgical techniques, the disease continues to be associated with high mortality rates, prolonged hospital stays and significant healthcare costs. Therapeutic options consist of empiric, broad-spectrum antibiotics, fibrinolytics, and a technique to drain the cavity [8,9]. The mortality rate of PPEs requiring pleural drainage ranges from 7% to 10%, and that of empyema ranges from 14% to 20% [2].

Few studies have analyzed the risk factors for developing CPPE, probably because the existing reports (conditioned by the low incidence of the disease) are generally small series. Thus, additional research is necessary to validate the results of previous studies and to propose lines of work that advance current knowledge. Although nowadays we have better diagnostic techniques and new treatments that can improve outcomes, predictive models are still necessary to prevent CPPE/empyema, optimize its treatment, and reduce complications. This study aims to identify factors associated with the development of CPPE/empyema in hospitalized patients with CAP, to assess the effect of corticosteroids in CAP patients, and to build a predictive model of CPPE/empyema in patients with pneumonia.

## 2. Materials and Methods

### 2.1. Study Design and Population

This is an observational case–control study nested within a retrospective cohort, based on clinical practice, and including all patients over 18 years of age who were hospitalized with a diagnosis of CAP from January 2009 to December 2019. CAP was defined as the presence of compatible symptoms (cough, expectoration, pleuritic pain, fever) together with evidence of a recent infiltrate on imaging test (chest X-ray, ultrasound, computed tomography [CT]) in a person who was not hospitalized, had not been hospitalized in the 10 days prior to the onset of symptoms, or was hospitalized but presented acute infection in the 24–48 h following admission. The hospital discharge database was searched for diagnoses coded as CAP (International Classification of Diseases, 10th revision [ICD-10] J18.9), PPE (ICD-10 J90) and empyema (ICD-10 J86.9), and 4372 consecutive medical records were reviewed. Patients presenting CAP and PPE or blunting of the costophrenic angle who did not undergo thoracentesis, or who had heart failure (HF), were excluded. Patients were also excluded if they had pediatric pneumonia or CPPE/empyema, healthcare-associated, or viral pneumonia; active neoplasia; chronic pleural pathology, tuberculosis, or opportunistic pathogen infections; were being treated with immunosuppressants, including systemic corticosteroids for more than 30 days, methotrexate, azathioprine, or anti-TNF; had a history of solid organ transplant or empyema; or were in palliative care. Patients with chronic obstructive pulmonary disease (COPD) under treatment with inhaled corticosteroids and HIV patients with a CD4 count of over 350 cells/mm^3^ and good virological control (not meeting criteria for AIDS) were not excluded.

Microbiological studies included antigen determinations for *Legionella* and pneumococcus in urine, Gram stain and sputum culture in patients from whom sputum of good quality could be obtained, and blood cultures in most patients presenting with fever on admission. All pleural fluids were cultured. During the epidemic period, PCR for influenza and respiratory syncytial virus (RSV) was performed. Since the study is retrospective and covers the period from 2009 to 2019, syndromic PCR results of respiratory samples were not available for the entire period and were therefore not included.

Cases and controls were defined according to the development (or not) of CPPE/empyema during admission in patients hospitalized for CAP. The date of diagnosis of CPPE/empyema was defined as the date of the thoracentesis. For each case, two controls were randomly selected and matched for the period of admission to avoid seasonality bias.

A chest X-ray was performed in all patients within 24 h of admission. If indicated, the patient underwent a second X-ray, CT scan, or thoracic ultrasound. Thoracentesis was performed in all patients with pleural effusion unless the procedure was considered unsafe, the effusion was small, or the patient had HF. The pleural fluid was analyzed to determine pH, proteins, LDH, glucose, Gram stain, culture, and cytology.

### 2.2. Variables

Baseline explanatory variables were demographic characteristics (age and sex); comorbidities (heart disease, chronic obstructive pulmonary disease (COPD), liver disease, kidney disease, cerebrovascular disease, and psychiatric disease); toxic habits (smoking, alcohol intake, and parenteral drug use); and antibiotic treatment in the previous three months.

Clinical-analytical variables comprised symptoms on presentation to the emergency department (ED), including cough, expectoration, dyspnea, chest pain, and sepsis criteria (as the inclusion of patients began in 2009, the 2012 Surviving Sepsis Campaign criteria were applied [10]). Data from the physical examination in the ED were also collected: systolic and diastolic blood pressure (BP), temperature, heart rate, and O_2_ saturation. Variables from complementary examinations included serum hemoglobin, leukocytes, platelets, C-reactive protein (CRP), procalcitonin (PCT), blood glucose, urea, creatinine, albumin, sodium, potassium, aspartate aminotransferase (AST), alanine aminotransferase (ALT), and arterial blood gases (ABG), including pH, partial pressure of oxygen (PaO_2_), partial pressure of carbon dioxide (PCO_2_), bicarbonate (HCO_3_), and lactate. Values were considered abnormal if they were above the reference values, except in the case of albumin and hemoglobin, which were considered abnormal if they were below, and sodium, potassium, leukocyte, and platelet counts, which were considered abnormal both above and below the reference values.

Microbiological variables consisted of Legionella and pneumococcus antigens in urine, sputum culture and Gram stain, blood cultures, pleural fluid culture, and PCR for influenza and RSV (in epidemic periods).

Radiological variables were presence of pleural effusion, unilobar or multilobar pulmonary consolidation, and loculation of pleural effusion on chest X-ray and/or CT.

Other variables included the Charlson index, treatment with systemic corticosteroids on admission (either due to application of the hospital’s clinical pathway or at the discretion of the attending clinician), CURB-65 [11], and FINE/PSI [12] scales, the American Thoracic Society (ATS) CAP severity scale [13], and the pre-treatment interval (days between symptoms onset and presentation to the ED).

### 2.3. Ethics

The ethics and clinical trials committee of Elda General University Hospital approved the study (no. PI2019/17), which was performed under conditions of real-world clinical practice.

### 2.4. Sample Size

A multivariable model was necessary to identify factors associated with the development of CPPE/empyema. With the obtained sample size of 318 patients (106 cases and 212 controls), a multivariable logistic regression model could be fitted with a maximum of 10 variables, according to the rule of 10 cases for each variable in the model [14].

### 2.5. Statistical Analysis

Imputation was performed using the mean value for analytical variables with less than 1% missingness (systolic and diastolic BP, heart rate, glucose, urea, creatinine, potassium, and ALT) and for alcohol intake, due to its greater frequency. The analytical variables were categorized as normal or abnormal, with a third category created for variables with more than 1% missingness.

A descriptive analysis of all variables was performed by calculating frequencies for categorical variables, and range, mean, and standard deviation for quantitative ones. The normality of the quantitative variables was verified using the Kolmogorov-Smirnov test.

Factors associated with the presence of CPPE/empyema were analyzed using contingency tables, applying the chi-squared test for categorical variables and the student’s *t*-test for quantitative variables. To estimate the magnitude of the associations, multivariable logistic regression models were fitted. Results were expressed as odds ratios (OR) and their 95% confidence intervals (CIs). A stepwise variable selection procedure was undertaken according to the Akaike information criterion. Indicators of goodness-of-fit and discriminative capacity are shown using the results of the likelihood ratio test (LRT) and the area under the receiver operating characteristic curve (AUC), with their 95% CIs. An internal bootstrap validation was performed using 200 simulations, obtaining the honest AUC with its 95% CI. All analyses were performed with SPSS (v.28) and R (v.4.0.2) software.

## 3. Results

Of the 4372 patients admitted with CAP from 2009 to 2019, 2015 were excluded because no thoracentesis was performed, as it was considered unsafe, the effusion was small, or because the effusion was considered secondary to HF (Figure 1). Of the 2357 remaining patients with CAP, 106 developed CPPE/empyema and were considered cases, and 212 controls with CAP but without CPPE/empyema were selected and matched with cases for the admission period (Figure 1).

Table 2 and Table 3 compare cases and controls according to different explanatory variables. The sample was predominantly made up of men (68.2%), with no significant differences between patients with and without CPPE/empyema. In the crude analysis, variables associated with CPPE/empyema were alcohol intake (drinker/ex-drinker) (*p* = 0.041), heart disease (*p <* 0.001), pleuritic pain (*p <* 0.001), radiological involvement (*p <* 0.001), treatment with corticosteroids on admission (*p <* 0.001), CURB-65 score (*p <* 0.001), heart rate (*p* = 0.005), leukocyte count (*p <* 0.001), and platelet count (*p <* 0.001). Results of the microbiological analyses are shown in Table 4.

Patients with CPPE/empyema were younger (*p* < 0.001) and had less comorbidity (*p* = 0.003) and a longer pre-treatment interval (*p* < 0.001) than controls without CPPE/empyema (Table 5).

Six variables (pre-treatment interval, Charlson index, pleuritic pain, radiological involvement, leukocytes, and treatment with corticosteroids) were included in the multivariable model, as these showed the most significance in the crude model and had no missing values.

Those that were independently associated with CPPE/empyema were pleuritic pain (OR7.42), multilobar radiological involvement (OR4.48), and leukocytosis (OR4.12). Corticosteroids showed a protective effect (OR 0.24). Age (*p* = 0.56) and sex (*p* = 0.074) acted as adjustment variables (Table 6).

This model obtained an AUC of 0.889 (95% CI 0.852–0.927, *p <* 0.001). After bootstrapping validation with 200 simulations, the AUC was 0.847 (95% CI 0.772–0.921). The discriminative capacity of the multivariable model with the six included variables presents moderate/high accuracy.

The equation of the multivariable logistic model is as follows:Probability of having CPPE/empyema = 1/1 + A
where A = exp[3.5256 − 0.6466 × sex + 0.0082 × age − 0.0759 × pre-treatment interval − 0.0064 × Charlson index − 2.0043 × pleuritic pain − 1.5001 × multilobar radiological involvement − 1.4164 × leukocytes + 1.4297 × corticosteroids].

The items in the equation are sex (1 = male, 0 = female), age (years), pre-treatment interval (days from symptoms onset to presentation to the ED), Charlson index (Charlson index score), pleuritic pain (1 = present, 0 = absent), radiological involvement (1 = multilobar, 0 = unilobar), leukocytes (blood leukocytes: 0 = normal [3800–11,500], 1 = abnormal [<3800 or >11,500], and corticosteroids (treatment with corticosteroids on admission: 1 = yes, 0 = no).

A probability cutoff point can be chosen to classify a new patient admitted for CAP as “will have CPPE/empyema” or “will not have CPPE/empyema” during hospital admission. Table 7 and Table 8 present the different cutoff points to estimate the probability in the model, along with their 95% validity indicators (sensitivity and specificity, Youden index, accuracy rate, positive and negative predictive values, and the probability ratios) for each cutoff.

According to the Youden index, the cutoff point that maximizes sensitivity and specificity is 0.50. Another cutoff of interest may be 0.30, which yields a sensitivity and specificity of nearly 80%. For the model to classify a new patient as having empyema during hospital admission, the model’s probability must be equal to or greater than the chosen cutoff.

## 4. Discussion

Our study identified five variables associated with an increased risk of developing CPPE/empyema in hospitalized patients with CAP: the length of the pre-treatment interval (that is, the days from symptoms onset to presentation to the ED), the Charlson index, pleuritic pain, multilobar radiological involvement, and an abnormal leukocyte count. The administration of systemic corticosteroids on admission was associated with a decreased risk. The discriminative capacity of the obtained multivariable model presents moderate/high accuracy for the development of CPPE/empyema.

Of the 2357 pneumonias included in the study, 106 (4.5%) developed CPPE/empyema. Some studies [7,8,15] have reported an increase in the incidence of CPPE/empyema in patients with CAP, possibly related to improvements in diagnostic methods and clinical detection and to the greater longevity of the population. Falguera et al. [15] analyzed 4715 CAPs in two hospitals over a 13-year period (February 1996 to December 2008), finding 261 patients (5.5%) who developed CPPE/empyema. Chalmers et al. [16] reported an even higher rate, with 92 out of 1269 (7.2%) included patients developing CPPE/empyema. The lower percentage observed in our sample may be due to improvements in both the diagnosis and treatment of CAP in the last decade, as our study period is more recent than those of the studies by the Falguera [15] and Chalmers [16]. The administration of corticosteroids upon admission may have also played a protective role. On the other hand, some of the excluded patients might have had complicated effusions, so we cannot rule out some underestimation of the cumulative incidence. That said, in the patients who did not undergo thoracentesis, the effusion was minimal, it was resolved with antibiotic treatment, or the patient had a clear HF diagnosis. In some patients, their baseline condition precluded the performance of a drainage procedure. The study by Falguera [15] took place in two reference hospitals, which may also explain the higher observed rates of CPPE/empyema in their cohort.

Most of our patients were younger men, and the cases that developed CPPE/empyema were younger and had fewer comorbidities than the controls, which is consistent with previous reports [15,16,17]. This finding may be attributable to a lower inflammatory response in older people. However, other studies report a higher incidence in people over 65 years of age [7]. Furthermore, like Falguera [15] and Chalmers [16], we observed a higher proportion of men compared to women who developed CPPE/empyema, although the reason for this apparent difference is not clear [18]. In general, the incidence of chest cavity infection is twice as high in men as in women [14]; however, the literature reviewed does not contain any pathophysiological explanation for this.

Some studies have observed a higher prevalence of diabetes, long-term excessive alcohol intake, injection drug use, and rheumatoid arthritis in patients with CPPE/empyema than in those with uncomplicated CAP [19,20,21]; however, our data do not corroborate these results

Pleuritic pain yielded a high OR, which is logical since pleural inflammation produces pain and is sometimes the first sign that the patient may be developing CPPE/empyema. In Falguera’s study [15], the presence of pain was also significant. Unlike other authors [15,16,19,20,21], we did not observe a significant association with alcohol intake, parenteral drug use, COPD, or hyponatremia. A retrospective study identified several risk factors for the development of pleural effusion in elderly patients with CAP, including low serum albumin and low blood sodium levels [22]. Albumin showed a significant association in our study, but it was excluded from the model due to the high rate of missingness. Abnormalities in leukocytes, increased CRP, and platelets may also reflect a greater degree of inflammation than in uncomplicated CAP. Multilobar involvement has not been assessed in other studies, but it is evident that it may also be related to greater severity.

Although PCR for influenza and RSV were performed during the epidemic period, no viral pneumonias were included. The study is retrospective and covers up to 2019, a period predating the COVID-19 pandemic, so no patients with COVID pneumonia were included. During the period of the COVID pandemic, no CPPE/empyema was recorded in the hospitals included in the study. No patient with COVID pneumonia developed CPPE/empyema.

Our study confirms some of the results of previous studies in highlighting the importance of clinical parameters like pleuritic pain and laboratory parameters such as CRP, leukocytes, and platelets [15,16]. The novelty of the study is the protective role that corticosteroids seem to play and the association with multilobar involvement in CAP.

Prior use of inhaled glucocorticoids for COPD or asthma has been associated with a lower incidence of CPPE [23]. The reason for this inverse association is unknown, although an abnormal inflammatory response to inhaled glucocorticoidsis one possible explanation. We did not observe an association with inhaled corticosteroids, but systemic corticosteroids on admission were associated with a lower presence of CPPE/empyema in our sample, possibly because of the decreased inflammatory response at the pleural level. Previous studies have not analyzed this question [15,16], and in fact, the use of corticosteroids in CAP is controversial. The Infectious Diseases Society of America [24] recommend them only in patients with septic shock refractory to vasoactive drugs, while European guidelines propose their use in patients with shock in general [25]. A recent phase III clinical trial [26] assessed the effects of hydrocortisone versus placebo on 28-day mortality in patients admitted to the intensive care unit with severe CAP but without septic shock. Patients receiving hydrocortisone had a lower risk of death by day 28 than those receiving placebo; however, the development of pleural effusion was not among the outcomes evaluated.

In a small, multicenter, double-blind, randomized placebo-controlled pilot study (STOPPE) [27], patients with CAP and pleural effusion were randomized (2:1) to receive either intravenous dexamethasone (4 mg twice daily for 48 h) or placebo, with 30-day follow-up. Investigators assessed a wide range of clinical, serological, and imaging outcomes, but they did not find any preliminary benefit for systemic corticosteroids in adults with parapneumonic effusions.

In our study, a high proportion of patients were treated with corticosteroids from admission, either due to bronchospasm or because this treatment is included in our hospital’s clinical pathway for CAP, specifically in patients with FINE > IV or CURB-65 > 3 at a dose of 0.5 mg/kg of methylprednisolone every 12 h. This early treatment may have promoted an early anti-inflammatory effect and protected against the development of pleural effusion. In the STOPPE study [27,28], corticosteroids were administered to patients who had already had a pleural effusion, and their effect on different clinical parameters was then assessed. Our results therefore raise interesting possibilities, although a prospective multicenter study would be necessary to verify the protective effect.

The discriminative capacity of our multivariable model presents a moderate/high AUC. The usefulness of this model, together with the calculation of the probability of developing CPPE/empyema through the logistic equation and the calculations of different accuracy measures for each possible cutoff, is of great relevance for clinical practice. According to the interpretation of these parameters, a cutoff of 0.50 maximizes the sensitivity and specificity of the predictive model for CPPE/empyema.

In practice, if a patient is admitted with CAP, the variables of the obtained model would be measured and entered into the equation, and the patient would then be classified as being at high or low risk for developing CPPE/empyema. The clinician could contrast each cutoff point with the full range of measures of predictive accuracy.

Although thoracentesis is still necessary in suspected CPPE/empyema, both for the correct diagnosis and for the drainage decision, the model presented here can help select patients at high risk of CPPE/empyema, who should be monitored more closely. This group of patients could benefit from initial treatment with corticosteroids.

Our study has some limitations. It took place in just two hospitals, so the results need to be corroborated in other contexts. However, they may be useful in hospitals similar to ours. As the study was retrospective, data collection was limited to that available in the medical records, some of which may have been missing. Nevertheless, all the variables included in the multivariable model were present before the onset of CPPE/empyema. In addition, we included only patients with CAP who were admitted to the hospital due to the impossibility of collecting all the data in those treated on an outpatient basis. This decision could result in an underestimation of the incidence. Patients with effusions but without thoracentesis were also excluded, which may entail a selection bias. The risk of selection bias in controls was minimized by matching for the same admission period in which the cases were included.

In addition, our study did not aim to identify the microbiological etiology of CAP. Its retrospective nature meant that microbiological studies were not available for all patients. However, pleural fluid culture was performed in all cases, and the number of positive cultures in the different samples was low. In the case of CPPE/empyema, the negativity may be conditioned by previous antibiotic treatment. As the aim was to determine the predictive factors on admission, the timing of the cultures—obtained when the CPPD/empyema had already developed—represents a limitation. Despite this, variables from microbiological studies were initially introduced into the statistical analysis, without finding any significant results.

Studies with a prospective design are necessary to confirm the results from predictive models.

## 5. Conclusions

In our study, the profile of the patient admitted for CAP who develops CPPE/empyema is a person under 60 years of age, with pleuritic pain, CAP severity criteria, multilobar radiological involvement, a high leukocyte and platelet count, and elevated CRP. The administration of corticosteroids is associated with a lower presence of CPPE/empyema. The multivariable model obtained, which shows a moderate/high discriminative capacity, may be relevant for clinical practice when it comes to explaining the development of CPPE/empyema in patients admitted for CAP.

## Figures and Tables

**Figure 1 jcm-14-01739-f001:**
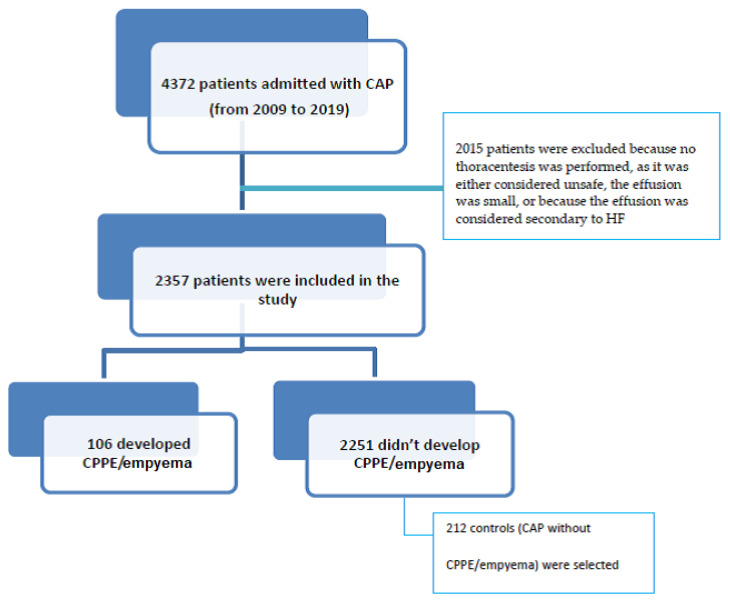
Patient flow chart.

**Table 1 jcm-14-01739-t001:** Staging of pleural effusion.

Stage 1: Simple or uncomplicated parapneumonic effusion	The fluid is free flowing and has exudative features with a protein content greater than 0.5 of the serum value and/or a lactate dehydrogenase (LDH) level greater than 0.6 that in serum (but usually <1000 international units [IU]/L). The leukocyte count is variable but neutrophils usually predominate. The fluid will have a normal pH and glucose level and there will be no evidence of infection by microorganisms.
Stage 2: Complicated parapneumonic Effusion and empyema	It is a fibrinopurulent stage, where bacterial invasion stimulates an inflammatory response resulting in fibrin deposition and loculations within the pleural space. The fluid characteristics are exudative with an elevated leukocyte count, pH < 7.20, glucose < 2.2 mmol/L (<40 mg/dL), and LDH > 1000 IU/L. If there is no pus, it is referred to as complicated parapneumonic effusion, but if there is frank pus, it is referred to as empyema.
Stage 3: Chronic organization	In the most advanced stage, the pleural fluid becomes organized, resulting in the appearance of a fibrous layer that envelops the lung, hindering complete lung expansion, impairing lung function, and increasing the possibility of further infection.

**Table 2 jcm-14-01739-t002:** Bivariable analysis of the association between CPPE/empyema and demographics, comorbidities, clinical and radiological variables, infectious origin, and corticosteroids treatment on admission.

Variables		Total N = 318		Controls (No CPPE/Empyema) N = 212		Cases (CPPE/Empyema) N = 106		*p* Value
		**N**	**%**	**n**	**%**	**n**	**%**	
Sex	Female	101	(31.8)	73	(72.3)	28	(27.7)	0.15
	Male	217	(68.2)	139	(64.1)	78	(35.9)	
Alcohol intake	Drinker/ex-drinker	273	(85.8)	188	(68.9)	85	(31.1)	0.041
	Never	45	(14.2)	24	(53.3)	21	(46.7)	
Heart disease	No	247	(77.7)	152	(61.5)	95	(38.5)	<0.001
	Yes	71	(22.3)	60	(84.5)	11	(15.5)	
COPD	No	265	(83.3)	173	(65.3)	92	(34.7)	0.24
	Yes	53	(16.7)	39	(73.6)	14	(26.4)	
Antibiotics in previous 3 months	No	265	(83.3)	182	(68.7)	83	(31.3)	0.089
	Yes	53	(16.7)	30	(56.6)	23	(43.4)	
Dyspnea	No	152	(47.8)	101	(66.4)	51	(33.6)	0.94
	Yes	166	(52.2)	111	(66.9)	55	(33.1)	
Cough	No	96	(30.2)	58	(60.4)	38	(39.6)	0.12
	Yes	222	(69.8)	154	(69.4)	68	(30.6)	
Expectoration	No	164	(51.6)	103	(62.8)	61	(37.2)	0.13
	Yes	154	(48.4)	109	(70.8)	45	(29.2)	
Pleuritic pain	No	192	(60.4)	163	(84.9)	29	(15.1)	<0.001
	Yes	126	(39.6)	49	(38.9)	77	(61.1)	
Sepsis	No	269	(84.6)	184	(68.4)	85	(31.6)	0.12
	Si	49	(15.4)	28	(57.1)	21	(42.9)	
Pleural effusion	No	210	(66.0)	210	(100)	0	(0.0)	<0.001
	Yes	108	(34.0)	2	(1.9)	106	(98.1)	
Radiological involvement	Unilobar	220	(69.2)	169	(76.8)	51	(23.2)	<0.001
	Multilobar	98	(30.8)	43	(43.9)	55	(56.1)	
Corticosteroids	No	254	(79.9)	156	(61.4)	98	(38.6)	<0.001
	Yes	64	(20.1)	56	(87.5)	8	(12.5)	

COPD: Chronic obstructive pulmonary disease, CPPE: complicated parapneumonic pleural effusion.

**Table 3 jcm-14-01739-t003:** Bivariable analysis of the association between CPPE/empyema and scales, physical examination parameters, and biological variables.

Variables		Total N = 318		Controls (No CPPE/Empyema) N = 212		Cases (CPPE/Empyema) N = 106		*p* Value
		n	(%col)	N	(%row)	n	(%row)	
CURB65	<2	118	(37.1)	67	(56.8)	51	(43.2)	<0.001
	≥2	174	(54.7)	135	(77.6)	39	(22.4)	
	Missing	26	8.2%	10	(38.5)	16	(61.5)	
FINE	<90	204	(64.2)	132	(64.7)	72	(35.3)	0.32
	≥90	114	(35.8)	80	(70.2)	34	(29.8)	
Systolic BP (mmHg)	Normal	134	(42.1)	82	(61.2)	52	(38.8)	0.17
	<120	119	(37.4)	82	(68.9)	37	(31.1)	
	>140	65	(20.4)	48	(73.8)	17	(26.2)	
Diastolic BP (mmHg)	Normal	224	(70.4)	143	(63.8)	81	(36.2)	0.15
	<60	39	(12.3)	31	(79.5)	8	(20.5)	
	>80	55	(17.3)	38	(69.1)	17	(30.9)	
Temperature (°C)	Normal	231	(72.6)	146	(63.2)	85	(36.8)	0.069
	<36	22	(6.9)	15	(68.2)	7	(31.8)	
	>38	65	(20.4)	51	(78.5)	14	(21.5)	
Heart rate (bpm)	Normal	219	(68.9)	157	(71.7)	62	(28.3)	0.005
	Abnormal	99	(31.1)	55	(55.6)	44	(44.4)	
O_2_ sat (%)	Normal	159	(50.0)	98	(61.6)	61	(38.4)	0.057
	Abnormal	159	(50.0)	114	(71.7)	45	(28.3)	
pH (ABG)	Normal	131	(41.2)	87	(66.4)	44	(33.6)	0.44
	Abnormal	132	(41.5)	92	(69.7)	40	(30.3)	
	Missing	55	(17.3)	33	(60.0)	22	(40.0)	
PaO_2_ (ABG) (mmHg)	Normal	33	(10.4)	20	(60.6)	13	(39.4)	0.31
	Abnormal	219	(68.6)	152	(69.4)	67	(30.6)	
	Missing	66	(20.8)	40	(60.6)	26	(39.4)	
PCO_2_ (ABG) (mmHg)	Normal	149	(46.9)	105	(70.5)	44	(29.5)	0.30
	Abnormal	115	(36.2)	75	(65.2)	40	(34.8)	
	Missing	54	(17.0)	32	(59.3)	22	(40.7)	
Bicarbonate (ABG) (mmol/L)	Normal	103	(32.4)	70	(68.0)	33	(32.0)	0.45
	Abnormal low (<22 mmol/L)	89	(28.0)	60	(67.4)	29	(32.6)	
	Abnormal high (>26 mmol/L)	69	(21.7)	49	(71.0)	20	(29.0)	
	Missing	57	(17.9)	33	(57.9)	24	(42.1)	
Lactate (mg/dL)	Normal	77	(24.3)	47	(61.0)	30	(39.0)	0.008
	Abnormal	45	(14.2)	39	(86.7)	6	(13.3)	
	Missing	195	(61.5)	125	(64.1)	70	(35.9)	
Glucose (mg/dL)	Normal	80	(25.2)	49	(61.3)	31	(38.8)	0.24
	Abnormal	238	(74.8)	163	(68.5)	75	(31.5)	
Urea (mg/dL)	Normal	189	(59.4)	121	(64.0)	68	(36.0)	0.23
	Abnormal	129	(40.6)	91	(70.5)	38	(29.5)	
Creatinine (mg/dL)	Normal	244	(76.7)	158	(64.8)	86	(35.2)	0.19
	Abnormal	74	(23.3)	54	(73.0)	20	(27.0)	
Albumin (g/dL)	Normal	41	(12.9)	34	(82.9)	7	(17.1)	<0.001
	Abnormal	22	(6.9)	3	(13.6)	19	(86.4)	
	Missing	254	(80.1)	174	(68.5)	80	(31.5)	
Sodium (mEq/L)	Normal	225	(70.8)	153	(68.0)	72	(32.0)	0.43
	Abnormal	93	(29.2)	59	(63.4)	34	(36.6)	
Potassium (mEq/L)	Normal	274	(86.2)	188	(68.6)	86	(31.4)	0.066
	Abnormal	44	(13.8)	24	(54.5)	20	(45.5)	
AST (IU/L)	Normal	235	(73.9)	160	(68.1)	75	(31.9)	0.47
	Abnormal	60	(18.9)	36	(60.0)	24	(40.0)	
	Missing	23	(7.2)	16	(69.6)	7	(30.4)	
ALT (IU/L)	Normal	235	(73.9)	158	(67.2)	77	(32.8)	0.72
	Abnormal	83	(26.1)	54	(65.1)	29	(34.9)	
Hemoglobin (g/dL)	Normal	210	(66.0)	146	(69.5)	64	(30.5)	0.13
	Abnormal	108	(34.0)	66	(61.1)	42	(38.9)	
Leukocytes (cells/µL)	Normal	111	(34.9)	96	(86.5)	15	(13.5)	<0.001
	Abnormal	207	(65.1)	116	(56.0)	91	(44.0)	
Platelets (cells/µL)	Normal	243	(76.4)	168	(69.1)	75	(30.9)	<0.001
	Abnormal low (<150,000)	40	(12.6)	31	(77.5)	9	(22.5)	
	Abnormal high (>450,000)	35	(11.0)	13	(37.1)	22	(62.9)	
CRP (mg/L)	Normal	10	(3.1)	8	(80.0)	2	(20.0)	0.51
	Abnormal	308	(96.9)	204	(66.2)	104	(33.8)	
PCT (ng/mL)	Normal	174	(54.7)	118	(67.8)	56	(32.2)	0.18
	Abnormal	55	(17.3)	31	(56.4)	24	(43.6)	
	Missing	89	(28.0)	63	(70.8)	26	(29.2)	

ABG: Arterial blood gases, ALT: alanine aminotransferase, AST: aspartate aminotransferase, BP: blood pressure, bpm: beats per minute, CPPE: complicated parapneumonic pleural effusion, CRP: C-reactive protein, PaO_2_: partial pressure of oxygen, PCO_2_: partial pressure of carbon dioxide, PCT: procalcitonin.

**Table 4 jcm-14-01739-t004:** Microbiological characteristics of cases and controls.

Microbiological Study	Cases (N = 106)	Controls (N = 212)	*p*
**Antigens in urine, n (%) performed**	**72 (67.9%)**	**164 (77.3%)**	
**Total**	**10 (13.9%)**	**13 (7.9%)**	
*Legionella*	0	1	0.67
Pneumococcus	10	12	0.10
**Blood tests, n (%) performed**	**66 (62.3%)**	**79 (37.3%)**	
Total	5 (7.6%)	8 (10.1%)	0.59
Coagulase-negative staphylococci	1		
*Streptococcus pneumoniae*	3	1	
*Streptococcus mitis*		1	
*Staphylococcus aureus*	1	1	
*Staphylococcus epidermidis*		1	
*Staphylococcus hominis*		1	
*Bacteroides*		1	
*Corynebacterium*		1	
*Enterococcus faecium*		1	
**Sputum cultures, n (%) performed**	**51 (48.1%)**	**75 (35.4%)**	
Total	10 (19.6%)	20 (26.7%)	0.36
*S. pneumoniae*	3	1	
*Haemophylus influenzeae*	2	5	
*Enterococcus aerogens*	1		
*Stenotrophomonas maltophilia*	1		
*Pseudomonas aeruginosa*	1	2	
*Serratia marcescens*	1		
*Moraxella catarrhalis*	1		
*S. aureus*		4	
*Klebsiella pneumoniae*		2	
*Escherichia coli*		1	
*Alcaligens*		1	
*Corynebacterium*		1	
*Shewanella*		1	
*Enterobacter cloacae*		1	
*Citrobacter youngae*		1	
**Pleural fluid culture, n (%) performed**	**106 (100%)**	**-**	
Total	33 (31.1%)		
*Streptococcus constellatus*	12		
*S. pneumoniae*	6		
*Streptococcus viridans*	2		
*Streptococcus pyogenes*	2		
*Streptococcus anginosus*	1		
*Staphylococcus intermedius*	1		
*Staphylococcus aureus*	3		
*Staphylococcus epidermidis*	2		
*E. cloaecae*	1		
*Gemella*	1		
*Peptoestreptococcus*	1		
*Fusobacterium nucleatum*	1		

**Table 5 jcm-14-01739-t005:** Bivariable analysis of quantitative variables.

Quantitative Variables	Controls (No CPPE/Empyema) N = 212		Cases (CPPE/Empyema) N = 106		*p* Value
	Mean	SD	Mean	SD	
Age in years	67.91	18.92	59.41	17.43	<0.001
Charlson index	3.62	2.33	2.76	2.64	0.003
Pre-treatment interval	5.50	7.56	13.17	14.78	<0.001

CPPE: complicated parapneumonic pleural effusion, SD: standard deviation.

**Table 6 jcm-14-01739-t006:** Variables associated with the development of CPPE/empyema according to the multivariable logistic regression model.

		**OR**	**95% CI**	***p*** **Value**
Male sex		1.91	(0.94–3.88)	0.074
Age		0.99	(0.97–1.02)	0.56
Pre-treatment interval		1.08	(1.04–1.12)	<0.001
Charlson index		1.01	(0.82–1.23)	0.95
Pleuritic pain		7.42	(3.83–14.38)	<0.001
Radiological involvement	Unilobar	1		
	Multilobar	4.48	(2.26–8.88)	<0.001
Leukocytes	Normal 3800–11,500	1		
	Abnormal	4.12	(1.94–8.76)	<0.001
Corticosteroid treatment	No	1		
	Yes	0.24	(0.09–0.61)	0.003

N = 318; CPPE/empyema = 106; LRT = 156.3 (*p* < 0.001); area under the ROC curve = 0.889 (95% CI 0.852–0.927).

**Table 7 jcm-14-01739-t007:** Measures of diagnostic accuracy at different cutoffs of the predictive model for complicated parapneumonic pleural effusion/empyema.

Cutoff	Sensitivity		Specificity		Youden Index	Total Accuracy %
	%	(95% CI)	%	(95% CI)		
0.05	98.1	(95.5–100.7)	32.5	(26.2–38.8)	0.306	54.4
0.10	96.2	(92.6–99.8)	47.2	(40.5–53.9)	0.434	63.5
0.15	93.4	(88.7–98.1)	59.9	(53.3–66.5)	0.533	71.1
0.20	89.6	(83.8–95.4)	70.3	(64.1–76.5)	0.599	76.7
0.25	84.0	(77.0–91.0)	75.5	(69.7–81.3)	0.595	78.3
0.30	79.2	(71.5–86.9)	80.2	(74.8–85.6)	0.594	79.9
0.35	77.4	(69.4–85.4)	83.0	(77.9–88.1)	0.604	81.1
0.40	74.5	(66.2–82.8)	85.4	(80.6–90.2)	0.599	81.8
0.45	72.6	(64.1–81.1)	88.7	(84.4–93.0)	0.613	83.3
**0.50**	**71.7**	**(63.1–80.3)**	**90.1**	**(86.1–94.1)**	**0.618**	**84.0**
0.55	68.9	(60.1–77.7)	92.0	(88.3–95.7)	0.609	84.3
0.60	61.3	(52.0–70.6)	95.3	(92.5–98.1)	0.566	84.0
0.65	55.7	(46.2–65.2)	95.8	(93.1–98.5)	0.515	82.4
0.70	50.0	(40.5–59.5)	96.7	(94.3–99.1)	0.467	81.1
0.75	44.3	(34.8–53.8)	96.7	(94.3–99.1)	0.410	79.2
0.80	38.7	(29.4–48.0)	97.2	(95.0–99.4)	0.359	77.7
0.85	30.2	(21.5–38.9)	97.6	(95.5–99.7)	0.278	75.2
0.90	17.0	(9.8–24.2)	98.6	(97.0–100.2)	0.156	71.4
0.95	8.5	(3.2–13.8)	99.5	(98.6–100.4)	0.080	69.2

**Table 8 jcm-14-01739-t008:** Predictive values and likelihood ratios of the predictive model for complicated parapneumonic pleural effusion/empyema.

Cutoff	PPV	(95% CI)	NPV	(95% CI)	LR+	(95% CI)	LR−	(95% CI)
0.05	42.1	(35.9–48.3)	97.2	(93.4–101.0)	1.4	(1.3–1.6)	0.1	(0.0–0.2)
0.10	47.7	(41.0–54.4)	96.2	(92.5–99.9)	1.8	(1.6–2.1)	0.1	(0.0–0.2)
0.15	53.8	(46.6–61.0)	94.8	(91.0–98.6)	2.3	(2.0–2.8)	0.1	(0.1–0.2)
0.20	60.1	(52.5–67.7)	93.1	(89.2–97.0)	3.0	(2.4–3.7)	0.1	(0.1–0.3)
0.25	63.1	(55.1–71.1)	90.4	(86.1–94.7)	3.4	(2.7–4.4)	0.2	(0.1–0.3)
0.30	66.7	(58.5–74.9)	88.5	(84.0–93.0)	4.0	(3.0–5.3)	0.3	(0.2–0.4)
0.35	69.5	(61.2–77.8)	88.0	(83.5–92.5)	4.5	(3.4–6.2)	0.3	(0.2–0.4)
0.40	71.8	(63.4–80.2)	87.0	(82.4–91.6)	5.1	(3.7–7.1)	0.3	(0.2–0.4)
0.45	76.2	(67.9–84.5)	86.6	(82.1–91.1)	6.4	(4.4–9.4)	0.3	(0.2–0.4)
**0.50**	**78.4**	**(70.2–86.6)**	**86.4**	**(81.9–90.9)**	**7.2**	**(4.8–10.9)**	**0.3**	**(0.2–0.4)**
0.55	81.1	(73.0–89.2)	85.5	(80.9–90.1)	8.6	(5.4–13.7)	0.3	(0.3–0.5)
0.60	86.7	(79.0–94.4)	83.1	(78.4–87.8)	13.0	(7.1–24.0)	0.4	(0.3–0.5)
0.65	86.8	(78.8–94.8)	81.2	(76.4–86.0)	13.3	(7.0–25.3)	0.5	(0.4–0.6)
0.70	88.3	(80.2–96.4)	79.5	(74.6–84.4)	15.2	(7.3–31.6)	0.5	(0.4–0.6)
0.75	87.0	(78.0–96.0)	77.7	(72.7–82.7)	13.4	(6.4–28.0)	0.6	(0.5–0.7)
0.80	87.2	(77.6–96.8)	76.0	(70.9–81.1)	13.8	(6.2–30.6)	0.6	(0.5–0.7)
0.85	86.5	(75.5–97.5)	73.7	(68.6–78.8)	12.6	(5.3–30.1)	0.7	(0.6–0.8)
0.90	85.7	(70.7–100.7)	70.4	(65.2–75.6)	12.1	(3.9–37.6)	0.8	(0.8–0.9)
0.95	90.0	(71.4–108.6)	68.5	(63.3–73.7)	17.0	(2.4–120.6)	0.9	(0.9–1.0)

PPV: Positive predictive value; NPV: negative predictive value; LR+: positive likelihood ratio; LR−: negative likelihood ratio.

## Data Availability

The original contributions presented in this study are included in the article. Further inquiries can be directed to the corresponding authors.

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
