# Peer review of "Factors Associated with Complicated Parapneumonic Pleural Effusion/Empyema in Patients with Community-Acquired Pneumonia: The EMPIR Study"

_jcm, 2025, doi:10.3390/jcm14051739_

Round 1
Reviewer 1 Report
Comments and Suggestions for Authors
In this article, “Factors associated with complicated parapneumonic pleural 2 effusion/empyema in patients with community-acquired 3 pneumonia. EMPIR Study” we can find a compilation of cases that provide novel and interesting perspectives on empyema risk factors.
The authors have achieved methodologically improvable study. The introduction and the aim are clear, but it should be necessary to have a deeper review of the available evidence and previous studies conducted. Its originality and the relevance of its themes make a perfectly publishable article but there are some changes to achieve a higher quality.
As a case-control study, it would be interesting to see the results broken down and stratified according to the pathologies and comorbidities previously present in the patient sample. This could be a differential factor to understand complications, for example in male patients (probably related to greater chronic pulmonary pathology and less functional reserve).
On the other hand, the etiology of the infection could be a determining factor and including it as a possible confounding factor in the main objectives could enrich the article.
Grammar and scientific language are developed correctly and need no modification. The tables and figures provide very enriching visual information. A part of it, a list of abbreviations should be recommended to facilitate the understanding of proposed acronyms.
The following changes are proposed to the article:
Minor changes:
- A list of abbreviations should be recommended.
- A more in-depth review of the current status of the pathology and related complications in the introduction to properly contextualize the results.
- Increasing the number of references and more recent works in the bibliography section will allow easier access to the most current evidence on the subject and is important for inclusion in quality journals such as this one.
Once the above changes have been rectified, the quality of the article will be a step above.
Author Response
In this article, “Factors associated with complicated parapneumonic pleural 2 effusion/empyema in patients with community-acquired 3 pneumonia. EMPIR Study” we can find a compilation of cases that provide novel and interesting perspectives on empyema risk factors.
The authors have achieved methodologically improvable study. The introduction and the aim are clear, but it should be necessary to have a deeper review of the available evidence and previous studies conducted. Its originality and the relevance of its themes make a perfectly publishable article but there are some changes to achieve a higher quality.
In the introduction we have tried to contextualize the work and its justification, not to make an exhaustive review of complicated parapneumonic pleural effusions.
We note the scarcity of studies, however, an additional comment on current studies is made, one on risk factors for the development of complicated parapneumonic pleural effusion (Analysis of clinical characteristics and risk factors of community‑acquired pneumonia complicated by parapneumonic pleural effusion in elderly patients Mingmei Zhong, Ruiqin Ni, Huizhen Zhang and Yangyang Sun Zhong et al. BMC Pulmonary Medicine (2023) 23:355) and another on the effect of corticosteroids (Medical treatment of pleural infection: antibiotic duration and corticosteroid usefulness. Vasileios Skouras1, FoteiniChatzivasiloglou ,MarianthiIliopoulou and TheofaniRimpa. Breathe 2023; 19: 230134. )
As a case-control study, it would be interesting to see the results broken down and stratified according to the pathologies and comorbidities previously present in the patient sample. This could be a differential factor to understand complications, for example in male patients (probably related to greater chronic pulmonary pathology and less functional reserve).
Initially, the different results were analyzed at a disaggregated level, but since the sample was small, the results were not significant.
On the other hand, the etiology of the infection could be a determining factor and including it as a possible confounding factor in the main objectives could enrich the article.
Since this was a retrospective study, an exhaustive etiological study was not performed, which was not the objective of the study. As already mentioned in the discussion, the number of cultures of the different samples is scarce (sputum culture, blood cultures...). Although all pleural effusions were cultured, their positivity was too low to obtain significant associations.
A table with positive microbiological isolations is included in the article.
Grammar and scientific language are developed correctly and need no modification. The tables and figures provide very enriching visual information. A part of it, a list of abbreviations should be recommended to facilitate the understanding of proposed acronyms.
The following changes are proposed to the article:
Minorchanges:
- A list of abbreviationsshould be recommended.
We have made the list of abbreviations as recommended, although they are indicated in the text and those appearing in the tables are explained at the foot of the table.
This table will be added in the appendix of tables and figures and at the end of the article.
LIST OF ABBREVIATIONS
|
|
CAP |
community-acquiredpneumonia |
PPE |
parapneumonic pleural effusion |
CPPE |
complicatedparapneumonic pleural effusion |
E |
empyema |
LDH |
lactatedehydrogenase |
CT |
computedtomography |
ICD |
International ClassificationofDiseases |
HF |
heartfailure |
COPD |
chronic obstructive pulmonarydisease |
HIV |
human immunodeficiency virus |
AIDS |
Acquired Immune Deficiency Syndrome |
RSV |
respiratorysyncytial virus |
ED |
emergencydepartment |
BP |
bloodpressure |
CRP |
C-reactive protein |
PCT |
procalcitonin |
AST |
aspartateaminotransferase |
ALT |
alanineaminotransferase |
ABG |
arterial blood gases |
PaO2 |
partialpressureofoxygen |
PCO2 |
partial pressure of carbon dioxide |
HCO3 |
bicarbonate |
ATS |
American ThoracicSociety |
OR |
odds ratios |
CIs |
confidenceintervals |
LRT |
likelihood ratio test |
AUC |
areaunderthe curve |
SD |
standard deviation |
PPV |
positive predictive value |
NPV |
negative predictive value |
LR+ |
positive likelihood ratio |
LR- |
negative likelihood ratio |
- A more in-depth review of the current status of the pathology and related complications in the introduction to properly contextualize the results.
A new search has been performed to review the current status of the pathology and related complications. The introduction has been updated to adequately contextualize the results.
- Increasing the number of references and more recent works in the bibliography section will allow easier access to the most current evidence on the subject and is important for inclusion in quality journals such as this one.
After a new exhaustive search of recent literature, we have found a new article on the risk factors associated with the development of parapneumonic pleural effusion and another study on the impact of systemic corticosteroids in the treatment of community-acquired pneumonia, which we have included in the discussion.
Reviewer 2 Report
Comments and Suggestions for Authors
The purpose of this case-control study was to investigate factors associated with complicated parapneumonic pleural effusion/empyema in patients with community-acquired pneumonia. Despite these limitations, the article makes a significant contribution to the understanding of CPPE/empyema risk factors in CAP patients.
The manuscript requires some improvements before it can be reconsidered for review.
The authors should review the entire manuscript to ensure that all acronyms are defined at their first use (e.g. PPE, CPPE).
Introduction:
Please, clarify the specific gaps in current knowledge that this study aims to address.
Materials and Methods:
Given the relatively small number of cases (106) compared to the initial cohort (4372), is the study adequately powered to detect all relevant associations?
What specific criteria were used to diagnose CPPE/empyema?
Were there any cases where the diagnosis was ambiguous, and if so, how were these handled?
Given that this is a single-center study, how generalizable are the findings to other populations or healthcare settings?
Discussion:
Provide more specific recommendations for future research and clinical practice.
Author Response
The purpose of this case-control study was to investigate factors associated with complicated parapneumonic pleural effusion/empyema in patients with community-acquired pneumonia. Despite these limitations, the article makes a significant contribution to the understanding of CPPE/empyema risk factors in CAP patients.
The manuscript requires some improvements before it can be reconsidered for review.
The authors should review the entire manuscript to ensure that all acronyms are defined at their first use (e.g. PPE, CPPE).
The manuscript has been revised to define acronyms in their first use.
Introduction:
Please, clarify the specific gaps in current knowledge that this study aims to address.
There are few studies that analyze the risk factors for developing complicated parapneumonic effusion and empyema. These studies are from the first decade of the XXI century, we believe it is necessary to validate these results and try to provide new knowledge; in this sense we find interesting the protective factor of corticoids that would deserve a clinical trial.
Materials and Methods:
Given the relatively small number of cases (106) compared to the initial cohort (4372), is the study adequately powered to detect all relevant associations?
Since the main objective of the study is to identify factors associated with the development of CPPE/empyema, it is necessary to identify factors with independent predictive capacity, and the appropriate method for this is through the adjustment of a multivariate model. The significant predictors of a multivariate model indicate that they explain the variability of the presence of CPPE/empyema 'independently' of the rest of the predictors, that is, they have their own predictive capacity on the response. However, the bivariate analysis of tables 2 and 3 does not ensure that the observed effects are free of confounding bias, and that the effect is an effect adjusted by other variables. For all these reasons, the multivariate model is the appropriate method to solve the main objective.
According to the literature (Harrell 2015), to adjust a multivariate model with binary response, a minimum of 10 events (CPEE/empyema) are needed for each variable included in the model. In this case, 106 cases are available, so up to 10 predictor variables can be included. This number is sufficient to establish a consistent predictive model, such as the one fitted, with 8 final variables. This is explained in the Methods section, in the calculation of the sample size.
What specific criteria were used to diagnose CPPE/empyema?
Specific criteria for diagnosing CPPE/empyema are described in the introduction.
We have added a table to better visualize the different stages of parapneumonic pleural effusion.
Table. STAGING OF PLEURAL EFFUSION
STAGE 1: SIMPLE OR UNCOMPLICATED PARAPNEUMONIC EFFUSION |
The fluid is free flowing and has exudative features with a protein content greater than 0.5 of the serum value and/or a lactate dehydrogenase (LDH) level greater than 0.6 that in serum (but usually <1000 international units [IU]/L). The leukocyte count is variable but neutrophils usually predominate. The fluid will have a normal pH and glucose level and there will be no evidence of infection by microorganisms.
|
STAGE 2: COMPLICATED PARAPNEUMONIC EFFUSION AND EMPYEMA |
It is a fibrinopurulent stage, where bacterial invasion stimulates an inflammatory response resulting in fibrin deposition and loculations within the pleural space. The fluid characteristics are exudative with an elevated leukocyte count, pH <7.20, glucose <2.2 mmol/L (<40 mg/dL), and LDH >1000 IU/L. If there is no pus, it is referred to as complicated parapneumonic effusion, but if there is frank pus, it is referred to as empyema.
|
STAGE 3: CHRONIC ORGANIZATION |
In the most advanced stage, the pleural fluid becomes organized, resulting in the appearance of a fibrous layer that envelops the lung, hindering complete lung expansion, impairing lung function and increasing the possibility of further infection. |
Were there any cases where the diagnosis was ambiguous, and if so, how were these handled?
Ambiguous cases were excluded from the study.
Given that this is a single-center study, how generalizable are the findings to other populations or healthcare settings?
Our hospital has 500 inpatient beds, serves a population of 200,000 patients in the province of Alicante (Spain), it could be extrapolated to centers in our environment with similar characteristics, since clinical practice is standardized.
Discussion:
Provide more specific recommendations for future research and clinical practice.
It would be interesting to perform a prospective study to confirm the protective effect of corticosteroids. Microbiological studies should also be included, adding molecular microbiological tests, to detail the etiology more precisely and its associations.

Round 2
Reviewer 2 Report
Comments and Suggestions for Authors
The revised manuscript along with the responses have addressed my concerns well.
I have no more comments.
Comments on the Quality of English Language
It is fair.